# Identification and Functional Analysis of CAD Gene Family in Pomegranate (*Punica granatum*)

**DOI:** 10.3390/genes14010026

**Published:** 2022-12-22

**Authors:** Lei Hu, Xuan Zhang, Huihui Ni, Fangyu Yuan, Shuiming Zhang

**Affiliations:** Department of Ornamental Horticulture, School of Horticulture, Anhui Agricultural University, Hefei 230036, China

**Keywords:** pomegranate, cinnamyl alcohol dehydrogenase, lignin, bioinformatics analysis, gene expression

## Abstract

[Objective] Cinnamyl alcohol dehydrogenase (CAD) is a key enzyme in lignin biosynthesis. The aim of this study was to identify *CAD* gene family members in pomegranate and its expression correlation with seed hardness. [Methods] Based on the reported *CAD* sequence of Arabidopsis, the *CAD* gene family of pomegranate was identified by homologous comparison, and then phylogenetic, molecular characterization, and expression profile analysis were performed. [Results] Pomegranate *CAD* gene family has 25 members, distributed on seven chromosomes of pomegranate. All pomegranate CAD proteins have similar physical and chemical properties. We divide the family into four groups based on evolutionary relationships. The member of group I, called bona fide *CAD*, was involved in lignin synthesis. Most of the members of group II were involved in stress resistance. The functions of groups III and IV need to be explored. We found four duplicated modes (whole genome duplication or segmental (WGD), tandem duplication (TD), dispersed duplication (DSD), proximal duplication (PD) in this family; TD (36%) had the largest number of them. We predicted that 20 cis-acting elements were involved in lignin synthesis, stress resistance, and response to various hormones. Gene expression profiles further demonstrated that the *PgCAD* gene family had multiple functions. [Conclusions] Pomegranate *CAD* gene family is involved in lignin synthesis of hard-seeded cultivar Hongyushizi and Baiyushizi, but its role in seed hardness of soft-seeded cultivar Tunisia needs to be further studied.

## 1. Introduction

Pomegranate (*Punica granatum* L.) originated in Iran, Afghanistan, and other Middle East regions. It grows well in tropical and subtropical regions [1,2]. This fruit has health benefits and medicinal properties. Studies have shown that pomegranate juice has high antioxidant activity [3,4], which is loved by consumers all over the world. Most of the pomegranate cultivars in the market have hard seeds and are increasingly unpopular with consumers [5,6], which encourages investigators and breeders to progress and breed soft-seed cultivars [7]. It takes a long time to breed a fruit crop, so identifying relevant genes to help speed up the breeding process is an important task [8].

The seed coat of pomegranate contains high levels of lignin [9]. Some studies have pointed out that there was a positive correlation between pomegranate seed hardness and seed-coat lignin content [10,11], and that the soft-seeded pomegranate had lower lignin content, while having a relatively higher amount of cellulose compared with the hard-seeded one [12]. Development of hardness in pomegranate seeds is regulated by multiple gene families. For example, according to the study of Luo et al., the expression of caffeic acid O-methyltransferase (*COMT*), caffeoyl Co A3-O-methyltransferase (*CCOMT*), phenylalanine ammonia-lyase (*PAL*), 4-coumarate: CoA ligase (*4CL*), cinnamoyl-CoA reductase (*CCR*), and cinnamyl alcohol dehydrogenase (*CAD*) genes in soft-seeded pomegranates and hard-seeded pomegranates were different [13]. Dai et al. found that the expressions of *COMT*, ferulate 5-hydroxylase (*F5H*), and hydroxycinnamoyl transferase (*HCT*) genes in soft-seeded hawthorn were significantly down-regulated compared with those in hard-seeded hawthorn [14]. In addition, many transcription factors are also involved in lignin synthesis. Xue et al. found that the expression patterns of transcription factors *NAC*, *MYB*, and *WRKY* were different in soft-seeded pomegranates and hard-seeded pomegranates [15]. Development of hardness in pomegranate seeds is a complex process. Some studies have pointed out that mdm-miR164e and mdm-miR172b targeted transcription factors NAC1, MYB, and WRKY. The miRNA-mRNA network regulated seed hardness by changing cell wall structure through a series of complex biological processes [16].

Lignin is a kind of phenolic polymer widely existing in plants, and its content is second only to cellulose, which is the second largest organic matter in plants. The synthesis of lignin can be divided into three pathways: shikimic acid pathway, phenylpropane metabolic pathway, and lignin synthesis specific pathway [17,18]. The synthesis of lignin monomers involves 11 enzyme families. Cinnamyl alcohol dehydrogenase (CAD) catalyzes the final step of monolignol biosynthesis. Its activity is dependent on zinc ions and NADPH, and it belongs to the middle-chain dehydrogenase/reductase (MDR) superfamily [19,20]. Thomas and Hans isolated CAD enzymes in spruce cambium and soybean cells and first determined CAD activity [21]. The *CAD* gene was first identified in the stem of tobacco [22]. Subsequently, *CAD* genes have been identified in various plants.

In plants, *CAD* genes have a high degree of homology and have developed into a large family. Phylogenetic analysis divides *CAD* genes into three major groups, the first of which was considered to be true *CAD* [23]. Nine *CAD* genes were found in Arabidopsis [24]. *AtCAD4* and *AtCAD5*, classified as group I, were mainly involved in the synthesis of coniferol and sinapyl alcohol [24,25,26], and several *CADs* of group II and III were mainly expressed in lignin [24]. There were 15 *CAD* genes in poplar genome, which could be divided into three categories, and the expression level of class I genes was higher in lignified tissues [27]. Twenty-six *CADs* were identified in pear and it was confirmed that *CAD2* plays a role in lignin synthesis in the stone cells of white pear [28] and sand pear [29,30]. A total of 14 and 24 *CAD* genes were identified in wild strawberry and cultivated strawberry, and *CAD* genes were related to strawberry firmness, and it may be involved in the formation of strawberry flavor traits [31]. There were eight *CAD* genes in *Phryma leptostachya*; its main members showed substrate preference [32]. A total of 31 and 25 *CAD* genes were identified in two mulberry cultivars (*Morus notabilis* and Fengchi) [33]; *CADs* exhibited various expression patterns in response to diverse stresses in mulberry [34]. CADs have many functions and are involved in lignin synthesis and plant defense response [20,25].

In recent years, analysis of *CAD* genes in pomegranate has been reported. Zarei et al. isolated a *CAD* gene from pomegranate and indicated that it was highly expressed in the hard-seed genotype [17]. Niu et al. studied the protein expression profiles of soft-seeded pomegranate and hard-seeded pomegranate, in which the expression of *CAD* genes showed strong correlations with their encoding proteins [12]. However, the study of *CAD* genes family in pomegranate has not been reported. The members of the *CAD* family of pomegranate were identified and analyzed for the first time in this paper. Specifically, we constructed a phylogenetic tree of the gene members of *PgCAD* family, and analyzed their protein properties, gene structure, motif, chromosome position, and gene duplication events. In addition, we also performed cis-acting element analysis on the promoter region of *PgCAD* genes. The transcriptome data of soft-seeded pomegranate ‘Tunisia’ and hard-seeded pomegranate ‘Hongyushizi’ and ‘Baiyushizi’ were used to draw the expression heat map of all identified *PgCAD* genes. This study provides important clues for the role of *CAD* genes in lignin synthesis.

## 2. Materials and Methods

### 2.1. Plant Material

The pomegranate cultivars selected in this study were Tunisia, Hongyushizi, and Baiyushizi, which were all 9-year-old pomegranate cultivars grown in Anhui Agricultural University experimental base. All pomegranate trees were of similar growth and grow robustly. Fruit samples were collected at 40, 80, and 120 days after flowering, with three biological replicates per cultivar, and a total of 27 samples were collected. Seeds and arils were immediately extracted, placed in liquid nitrogen, and finally frozen at −80 degrees Celsius in the laboratory.

### 2.2. RNA Extraction, Library Construction, and Sequencing

Total RNA was extracted from samples using a Trizol reagent kit (Invitrogen, Carlsbad, CA, USA) according to the manufacturer’s protocol. RNA quality was assessed on an Agilent 2100 Bioanalyzer (Agilent Technologies, Palo Alto, CA, USA) and checked using RNase free agarose gel electrophoresis. After total RNA was extracted, mRNA was enriched by Oligo(dT) beads, and prokaryotic mRNA was enriched by removing rRNA by a Ribo-ZeroTM Magnetic Kit (Epicentre, Madison, WI, USA). Then, the enriched mRNA was fragmented into short fragments using a fragmentation buffer and reversely transcribed into cDNA by using a NEBNext Ultra RNA Library Prep Kit for Illumina (NEB #7530, New England Biolabs, Ipswich, MA, USA). The purified double-stranded cDNA fragments were end repaired, a base added, and ligated to Illumina sequencing adapters. The ligation reaction was purified with the AMPure XP beads (1.0X). Ligated fragments were subjected to size selection by agarose gel electrophoresis and polymerase chain reaction (PCR) amplified. The resulting cDNA library was sequenced using Illumina Novaseq6000 by Gene Denovo Biotechnology Co. (Guangzhou, China).

### 2.3. Identification of CAD Gene Family Members and Physicochemical Properties Analysis in Pomegranate

To identify the gene members of the *CAD* family in the pomegranate genome, CAD protein sequences in *Arabidopsis thaliana* were downloaded from TAIR (https://www.arabidopsis.org/, accessed on 3 September 2022). The amino acid sequence of AtCAD family (AT2G21890.1, AT2G21730.1, AT3G19450.1, AT4G34230.1, AT4G37970.1, AT4G37980.1, AT4G37990.1, AT4G39330.1, AT1G72680.1) was used as the query sequence to align the pomegranate genome in NCBI BLASTp online (https://blast.ncbi.nlm.nih.gov/Blast.cgi, accessed on 3 September 2022) (Gap costs: Existence: 11, Extension: 1) and extracted the sequence with E-value < 1 × 10^−20^. We downloaded the zinc-binding dehydrogenase domain from the protein database Pfam (http://pfam.xfam.org/, accessed on 4 September 2022) and used hmmsearch of Hmmer (https://www.ebi.ac.uk/Tools/hmmer/search/hmmer, accessed on 4 September 2022) to validate our results. After verification, we finally obtained all the members of the *PgCAD* family.

In order to preliminarily understand these identified *CAD* genes, we used the ProtParam tool of Expasy (https://web.expasy.org/protparam/, accessed on 10 September 2022) to analyze the basic properties of proteins; WoLF PSORT (https://wolfpsort.hgc.jp/, accessed on 11 September 2022) was used to predict protein subcellular localization.

### 2.4. Phylogenetic Analysis

To understand the evolutionary relationship of PgCAD, we used ClustalW to perform multiple sequence alignment of 25 PgCAD and 9 AtCAD protein sequences together; this result was imported into MEGA11 [35]. The neighbor-joining (NJ) statistical method was used for phylogenetic analysis. The bootstrap value was set to 1000, other parameters were default, and the result was imported into ITOL (https://itol.embl.de/, accessed on 20 September 2022) for visualization [36].

### 2.5. Chromosomal Position, Duplication Mode, and Collinearity Analysis

We downloaded a gene annotation file from NCBI (https://www.ncbi.nlm.nih.gov accessed on 2 September 2022) and used TBtools (v1.098769) [37] software to display gene positions on chromosomes. MCScanX of TBtools was used to identify gene duplication patterns and collinearity analysis.

### 2.6. Gene Structure and Protein Motif Analysis

We extracted the structural information of genes in pomegranate genome annotation to draw the gene structure map. The online software MEME (https://meme-suite.org/meme/, accessed on 23 September 2022) was used to predict the motif of pomegranate protein sequence. The MEME parameter was set to ‘Select the number of motif = 15’, and the other parameters were default. The above information was used to draw images in TBtools.

### 2.7. Analysis of Promoter Cis-Acting Elements

We used the Gtf/GFF3 Sequences Extractor option in TBtools to obtain 2000 bp upstream sequences of *CAD* gene and submitted them to Plant CARE (http://bioinformatics.psb.ugent.be/webtools/plantcare/html/, accessed on 25 September 2022) for promoter cis-acting element analysis. Finally, the common functional components were visualized by TBtools.

### 2.8. Expression of CAD Gene in Pomegranate

The expression information of pomegranate *CAD* genes was extracted from transcriptome data, and the log2(FPKM + 1) calculation was performed using TBtools and the expression heat map was drawn.

## 3. Results

### 3.1. Physicochemical Properties of Pomegranate CAD Proteins

We identified 25 *CAD* genes from the pomegranate genome, and they were named based on their position on the chromosome. The results of ExPASy analysis (Table 1) showed that the number of aminos of protein encoded by CAD proteins ranged from 202 to 431, and the molecular weight ranged from 21,601.50 to 45,082.65; there was no significant difference in the number and molecular weight of amino acids. The isoelectric points of these proteins were more acidic except for CAD10 and CAD24, which were basic. The instability coefficients of proteins were all less than 40, which were stable proteins, and their aliphatic index ranged from 77.60 to 98.64. Among the 25 proteins, only CAD4, 7, 16, 18, 19, 24, and 25 were hydrophobins, while other CADs were hydrophilic proteins. According to the prediction results of subcellular localization, we can see that all CAD proteins are localized in the cytoplasm.

### 3.2. Phylogenetic Analysis of the CAD Gene Family in Pomegranate

To reveal the evolutionary relationships for PgCAD, we downloaded the CAD protein sequences of five species and constructed the phylogenetic tree together with the CAD protein of pomegranate. The 88 CAD proteins were grouped into four subgroups (Figure 1). Group I was considered to be bona fide CAD, and its members were associated with lignin synthesis. Therefore, we suggested that PgCAD7 in this group might be involved in lignin synthesis. Group II was divided into two subgroups, group II-a and group II-b. In Arabidopsis, AtCAD7 and AtCAD8 were associated with plant resistance [24,27]. Therefore, we hypothesized that most PgCAD gene members of group II were involved in plant resistance. Group III contained PgCAD4 and PgCAD19. Among this group in Arabidopsis, AtCAD1 did not show high catalytic activity for lignin synthesis. Thus, whether PgCAD4 and PgCAD19 catalyze lignin synthesis remains to be determined. Group V contained CADs from five species except Arabidopsis. In this group, OsCAD6 and PtoCAD3 were not involved in lignin synthesis. Therefore, the functions of PgCAD24 and PgCAD25 need to be explored. Group IV only had CADs of rice, which was consistent with a previous study [38].

### 3.3. Chromosomal Position, Duplication Mode, and Collinearity Analysis

Twenty-five *PgCAD* genes were distributed on seven chromosomes of pomegranate (Figure 2), and the highest number of *CAD* was located in Chr3, followed by Chr1 and Chr6; few *PgCAD* genes were localized in the rest of the chromosome. We identified three pairs of paralogous gene in the pomegranate, all of which were located at high gene density positions on the chromosome. In addition, we also analyzed the Ka/Ks ratio of *PgCAD* gene pairs (Table 2). The Ka/Ks ratio of all gene pairs was less than one, indicating that they had undergone strong purification selection and played an important role in the evolution of *CAD* genes.

In the collinearity analysis between pomegranates and other species (Figure 3), we found five orthologous gene pairs between *P. granatum* and *A. thaliana*, and eight orthologous gene pairs between *P. granatum* and *Eucalyptus grandis*, and it was clear that there was a higher collinearity in *CAD* gene families between pomegranate and *E. grandis*.

There were five different kinds of duplication events in genes: whole genome duplication or segmental (WGD), tandem duplication (TD), dispersed duplication (DSD), proximal duplication (PD), and singleton duplication (SD). We found four of these in *PgCADs* and no SD events (Table 3). TD (36%) accounted for the majority of *CAD* genes, followed by WGD (20%) and DSD (20%); PD (12%) was the least. In Chr3, which had a large number of *CAD* genes, TD was absolutely dominant. On the whole, TD was also the most important replication mode; WGD and DSD also played an important role in the expansion of *PgCADs.*

### 3.4. The Gene Structures and Protein Domains Analysis of the CAD Gene Family in Pomegranate

In order to explore the gene structure characteristics of *PgCAD*, the exon–intron analysis of these genes was performed (Figure 4b). In some studies, the organization of exon–intron in *CAD* was divided into three modes; we found that the genetic structure of *PgCAD* was similar.

The *CAD* family members of exon–intron pattern I have five exons, with the third exon being the longest; the members of exon–intron pattern II have five exons, with the fourth exon being the longest; the members of exon–intron pattern III have six exons, with the fourth exon being the longest [6]. In our study, most *PgCAD* gene structures in group II were consistent with pattern I, the *PgCAD* gene structure of group I (*PgCAD7, PgCAD18*) was consistent with pattern II, the members of groups III and IV were consistent with pattern III, and *PgCAD10* and *PgCAD13* did not fall into any of these three modes. Similar findings were found for *CAD* genes in other species [28,39].

We predicted 15 conserved motifs of PgCAD (Figure 4c), and the results showed that motif1-6 appeared in all PgCADs, while motif7 and 9 only appeared in group II; motif15 only appeared in group IV. Clearly, PgCADs had highly similar structures and motifs, but there were still differences among different groups. In addition, motif 5 contained the Zn1 binding site and Zn2 binding site, and motif 4 contained the NADPH binding site.

### 3.5. Promoter Analysis of Pomegranate CAD Genes

To investigate the potential function of the gene members of the *PgCAD* family, we analyzed the 2000 bp sequence upstream of the CDS and predicted 20 cis-acting elements (Figure 5). Seven cis-acting elements were associated with phytohormones, including methyl jasmonate (MeJA), auxin, abscisic acid, salicylic acid, and gibberellin. This suggested that *PgCAD* might be involved in the response to these hormones. We also found elements involved in stress response, such as drought and low temperatures. We identified seven light responsive elements that were present in most *CAD* genes. It indicated that the expression of these *PgCADs* might be regulated by light. In addition, we found a large number of MYB binding sites upstream of the *PgACDs* promoter, and we hypothesized that MYB regulated *PgCADs.* We found no significant regularity in the cis-acting elements of these 25 genes.

### 3.6. Expression of PgCAD genes

To investigate expression patterns of *CAD* genes in soft-seed and hard-seed cultivars, we obtained transcriptomic date from both cultivars of fruit (Figure 6a). We found that two *PgCAD* genes (*PgCAD2*, *PgCAD16*) were not expressed at all stages of pomegranate fruit development, suggesting that these *CAD* genes might play roles in other organs. Overall, *PgCAD* genes expression showed spatiotemporal specificity. The expression of fourteen *PgCAD* genes was higher at T1 (40 days after anthesis). The expression of these genes was maintained at low levels in T2 (80 days after flowering) and T3 (120 days after flowering). *PgCAD1*, *PgCAD4*, *PgCAD9*, and *PgCAD15* were highly expressed in T2 and T3. *PgCAD3* and *PgCAD25* showed high expression levels in all stages of seed development of Hongyushizi, and *PgCAD10* and *PgCAD24* showed higher expression levels in T2 and T3 stages of Tunisia.

In order to better reflect the differences in *CAD* gene expression among different pomegranate cultivars, we selected the data at T1 stage to draw a heat map (Figure 6b). For *PgCAD7* and *PgCAD18*, which might be related to lignin synthesis, their expression patterns were different. The expression level of *PgCAD7* was the highest in Tunisia and the lowest in Baiyushizi. The expression level of *PgCAD18* was the highest in Hongyushizi and the lowest in Baiyushizi. Hongyushizi had the highest lignin content, followed by Baiyushizi, and Tunisia has the lowest lignin content [40]. The expression pattern of *CAD* genes in Hongyushizi and Baiyushizi was basically consistent with the content of lignin in the two cultivars. However, the expression of *PgCAD7* and *PgCAD18* in soft-seed cultivar ‘Tunisia’ was higher than that in a hard-seed cultivar.

## 4. Discussion

In recent years, there is no lack of reports about the medical effect of pomegranate, which makes this healthy fruit even more popular with consumers [6]. At present, the pomegranate on the market is divided into soft-seed pomegranate and hard-seed pomegranate; hard-seed pomegranate grain is harder, which creates troubles with edibility, and in the market is not welcomed by consumers. Therefore, more and more attention has been paid to the research on the formation mechanism of the pomegranate inner seed-coat hardness.

The seed hardness of pomegranate is positively correlated with the lignin content. Reducing the lignin content can significantly reduce the seed hardness [39]. CAD catalyzes the last step of lignin synthesis and is an important indicator of lignin content. In Arabidopsis phylogeny, AtCAD4 and AtCAD5 were grouped as group 1, which were considered to be true CAD, and they were confirmed to catalyze lignin biosynthesis. The AtCAD members of group 2 were involved in biotic and abiotic stress responses, while the AtCAD members of group 3 were not involved in lignin biosynthesis [25,26,41]. Because the gene members of *CAD* family of Arabidopsis were studied earlier and more thoroughly, we used it as a reference for the study of the *CAD* gene family of pomegranate.

In this study, *CAD* genes in the whole genome of pomegranate were studied for the first time, and 25 *CAD* genes were identified. We divided the gene members of the *PgCAD* family into four groups by phylogenetic analysis. Group I had only two members, PgCAD7 and PgCAD18, which were presumed to be involved in lignin synthesis. Group II had 19 members, which might respond to biotic and abiotic stress. PgCAD4 and PgCAD19 were members of group III, and group V had two members (PgCAD24, PgCAD25) whose functions need to be further defined. Interestingly, the number of CAD protein families varied greatly between species. There were nine CADs in Arabidopsis [26], thirteen in sweet potato [42], seven in oil palm [43,44], fifty-seven in pear [28], forty-two in apple [38], and twenty-four in strawberry [31]. No matter how large the CAD protein family was in a given species, there was only one or two true CADs. This suggested that CADs of group I were relatively stable during evolution, and their functions might not have changed much.

The differences in gene duplication patterns help us to study gene amplification and gene function diversity. Qi et al. analyzed the mode of the gene members of *CAD* family duplication in several different species and pointed out that WGD and SD were key to its expansion [38]. However, this conclusion was limited to the rose family. There were four duplication modes of the gene members of the *PgCAD* family, among which tandem duplication accounts for the majority, but does not have the absolute dominant position. WGD, DSD, and PD also play important roles in the expansion of pomegranate *CAD* genes. Historically, many ancient polyploid events had occurred in angiosperms, which promoted the amplification of many stress-related genes [41]. Many *CAD* gene families of group II in different kinds of plants were significantly affected by WGD, which may explain why most of them were involved in stress resistance.

Cis-acting elements are specific DNA sequences connected in series with structural genes. It binds transcription factors to regulate gene transcription [45]. We analyzed cis-acting elements in the promoter region of the gene members of the *PgCAD* family. We found several cis-acting elements in response to plant hormones, including: MeJA, auxin, abscisic acid, salicylic acid, and gibberellin. Plant hormones are closely related to the stress resistance of plants. This finding suggests that the gene members of the *CAD* family may be involved in plant stress response. In addition, we also found cis-acting elements in response to low temperature and drought. Much evidence points to the *PgCAD* gene being involved in plant stress response. More importantly, we found a MYB binding site in the promoter region of the *CAD* gene. MYB is a large family of transcription factors that regulate a variety of physiological processes, including lignin synthesis, plant stress tolerance, anthocyanin synthesis, etc. [46] This finding suggests that *PgCAD* may have multiple functions.

By analyzing the expression of the gene member of the *PgCAD* family, we found that the expression pattern of *PgCAD* showed obvious spatiotemporal specificity. Most *PgCAD* genes were highly expressed at T1 (including true group 1), and then decreased rapidly. Some genes were highly expressed at T2 and T3. This reflects the rich functions of the gene members of the *PgCAD* family. By analyzing the expression patterns of different pomegranate cultivars, we found that the expression levels of *PgCAD7* and *PgCAD18* were higher in soft-seed cultivar Tunisia, which was different from the variation trend of lignin content, Niu et al. also found a similar phenomenon [12]. The expression patterns of *PgCAD7* and *PgCAD18* in Hongyushizi and Baiyushizi were basically consistent with the lignin content between them; they may catalyze their lignin synthesis. However, functions of *PgCAD7* and *PgCAD18* in soft-seed cultivar ‘Tunisia’ are still unclear and need to be further explored.

## 5. Conclusions

We identified 25 *CAD* genes from the pomegranate genome and analyzed their phylogenetic relationships, duplication patterns, gene structures, conserved motifs, and cis-acting elements. The results showed that the *PgCAD* gene family may be involved in a variety of biological responses, group I may be involved in lignin synthesis, and group II may be involved in stress response. The gene expression profiles indicated that *PgCAD* genes expression showed spatiotemporal specificity. *PgCAD7* and *PgCAD18* showed a correlation with lignin synthesis in Hongyushizi and Baiyushizi, but not in Tunisia. Further study of their mechanisms of action in Tunisia may be helpful for breed improvement.

## Figures and Tables

**Figure 1 genes-14-00026-f001:**
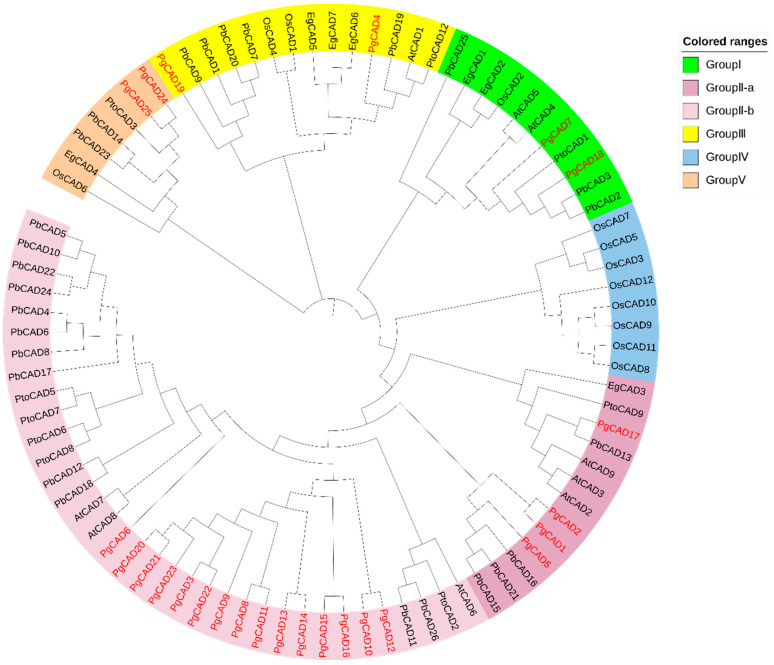
Phylogenetic relationships of cinnamyl alcohol dehydrogenase (CAD) proteins in pomegranate. The color blocks on the **right** indicate the different groups, and the *PgCADs* are highlighted in red.

**Figure 2 genes-14-00026-f002:**
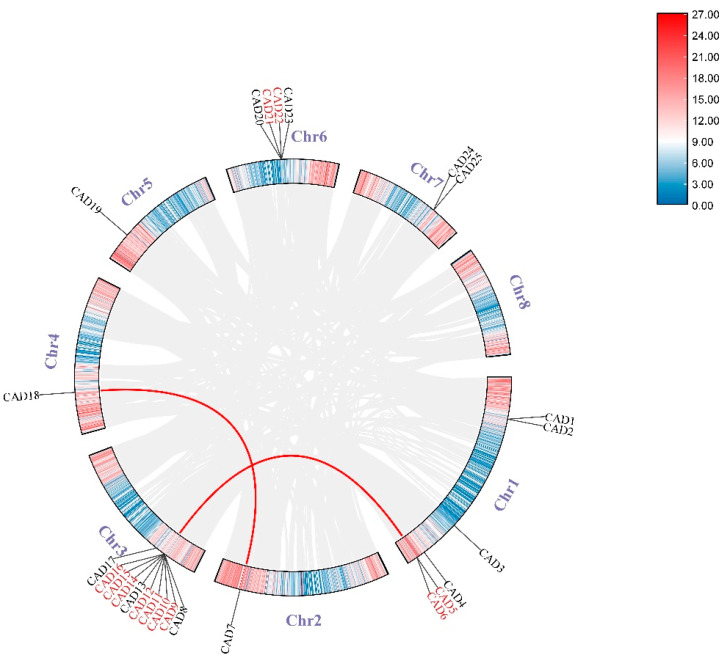
Chromosomal distribution of *PgCAD* genes. Chromosome numbers are shown next to each chromosome, and tandem duplicated genes are highlighted in red. The color bands on the chromosomes indicate gene density, with red indicating high gene density and blue indicating low gene density. The red line indicates three pairs of paralogous genes: *PgCAD5* and *PgCAD8*, *PgCAD6* and *PgCAD9*, *PgCAD7* and *PgCAD18*.

**Figure 3 genes-14-00026-f003:**
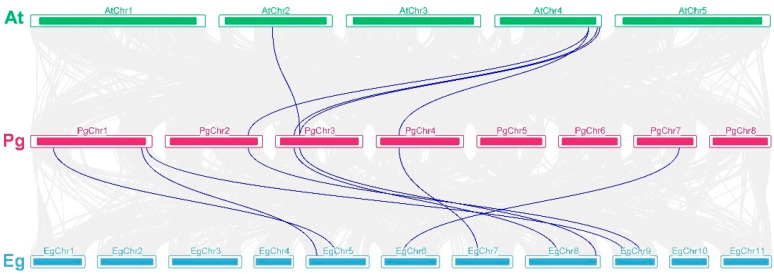
Genome-wide synteny analysis for *CAD* genes among *P. granatum*, *E. grandis*, and *A. thaliana*. The blue lines indicate ortholgs gene pairs. Eg, *E. grandis*; Pg, *P. granatum*; At, *A. thaliana*.

**Figure 4 genes-14-00026-f004:**
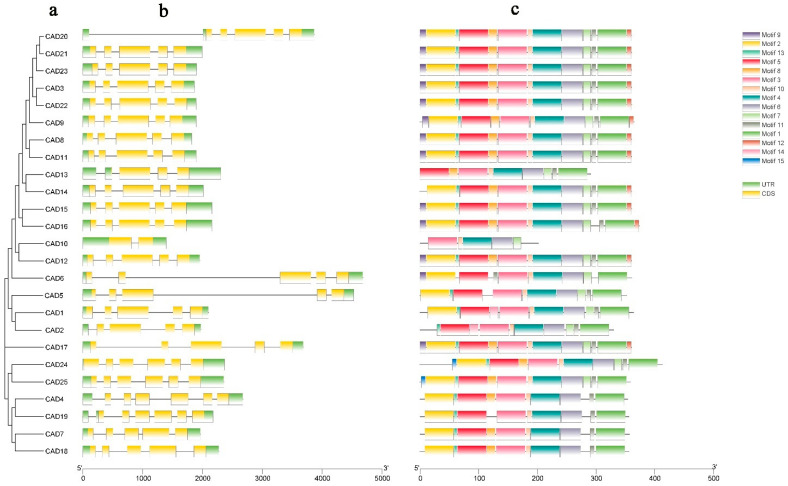
The gene structures and protein domains of *PgCAD* members. (**a**) Maximum likelihood phylogenetic tree of *PgCADs*; (**b**) Gene structures of *PgCADs*. Green boxes indicated UTR, yellow boxes indicated CDS, and gray lines indicated introns. UTR: untranslated region, CDS: coding sequence; (**c**) Protein motif of *PgCADs*. They were named according to the E-value of the motif. The scale at the bottom indicates the sequence length.

**Figure 5 genes-14-00026-f005:**
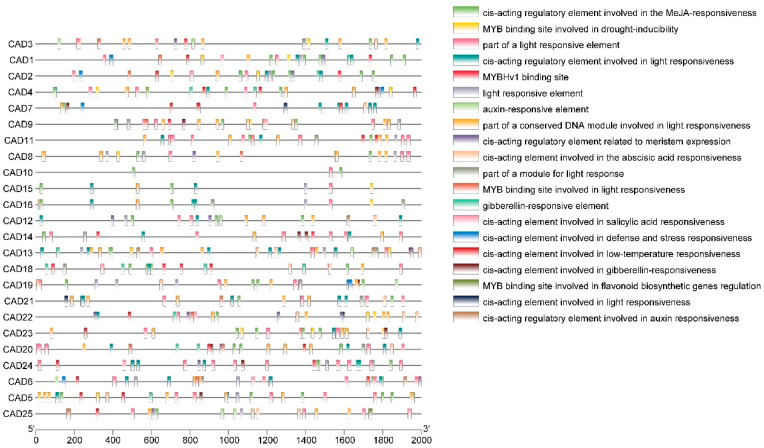
Cis-acting elements in promoter regions of *PgCAD*. The color blocks on the **right** indicate the different cis-acting elements. The scale at the bottom indicates the sequence length.

**Figure 6 genes-14-00026-f006:**
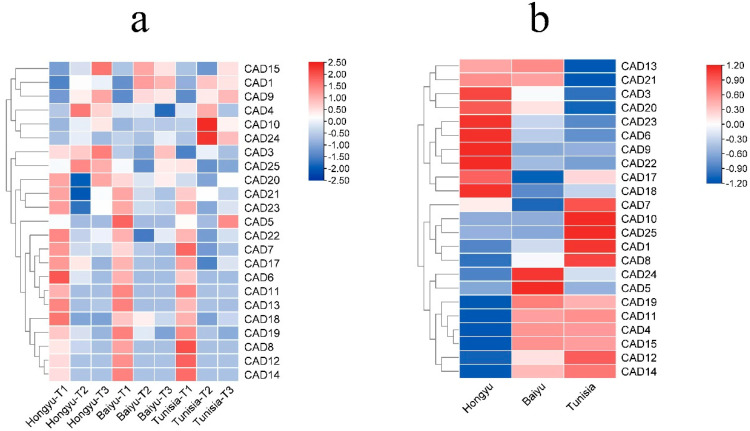
(**a**) Expression patterns of *PgCAD*. T1-T3 correspond to 40 days after anthesis, 80 days after anthesis, and 120 days after anthesis. (**b**) Expression patterns of different cultivars at T1. The scale on the **left** of the picture represents expression levels, with red indicating high expression and green indicating low expression.

**Table 1 genes-14-00026-t001:** The predicated protein information of *PgCAD*.

Name	Gene ID	No. of Amino Acide	Mol. Wt (kDa)	Isoelectric Point (pI)	Instability Index	Aliphatic Index	Grand Average of Hydropathicity	Subcellular Localization
PgCAD1	XM_031521130.1	364	39,091.80	5.94	31.86	91.04	0.008	Cytoplasm
PgCAD2	XM_031521131.1	330	35,401.70	5.70	29.57	94.21	0.050	Cytoplasm
PgCAD3	XM_031518621.1	361	39,241.30	5.73	32.26	87.76	0.023	Cytoplasm
PgCAD4	XM_031521510.1	354	39,203.86	7.54	38.20	77.60	−0.244	Cytoplasm
PgCAD5	XM_031549529.1	352	37,611.48	6.21	27.43	98.64	0.168	Cytoplasm
PgCAD6	XM_031549518.1	361	38,767.91	6.67	32.29	96.62	0.096	Cytoplasm
PgCAD7	XM_031526044.1	357	38,864.78	5.46	22.55	90.28	−0.003	Cytoplasm
PgCAD8	XM_031530115.1	361	38,658.47	5.95	28.74	92.02	0.066	Cytoplasm
PgCAD9	XM_031530113.1	365	39,148.38	6.70	29.78	89.70	0.043	Cytoplasm
PgCAD10	XM_031530121.1	202	21,601.50	7.78	28.40	114.41	0.279	Cytoplasm
PgCAD11	XM_031530114.1	361	38,783.69	6.06	27.40	92.02	0.044	Cytoplasm
PgCAD12	XM_031532959.1	361	38,792.61	5.70	38.79	92.88	0.015	Cytoplasm
PgCAD13	XM_031532961.1	291	31,297.27	6.14	24.01	95.05	0.084	Cytoplasm
PgCAD14	XM_031532960.1	361	38,892.13	6.31	23.22	93.91	0.099	Cytoplasm
PgCAD15	XM_031532956.1	374	40,503.65	6.31	28.21	92.46	0.021	Cytoplasm
PgCAD16	XM_031532957.1	361	39,055.97	6.43	29.04	92.27	−0.025	Cytoplasm
PgCAD17	XM_031532280.1	361	39,130.21	6.79	29.49	91.47	0.005	Cytoplasm
PgCAD18	XM_031536024.1	356	38,861.85	5.72	29.12	91.91	−0.055	Cytoplasm
PgCAD19	XM_031540038.1	356	38,754.64	5.65	26.95	84.86	−0.027	Cytoplasm
PgCAD20	XM_031545715.1	361	39,273.56	6.46	25.15	97.45	0.023	Cytoplasm
PgCAD21	XM_031544474.1	361	39,245.50	6.46	25.39	96.93	0.017	Cytoplasm
PgCAD22	XM_031544478.1	361	39,238.27	5.73	29.67	90.69	0.021	Cytoplasm
PgCAD23	XM_031544479.1	361	39,143.31	6.13	28.53	93.66	0.017	Cytoplasm
PgCAD24	XM_031548045.1	413	45,082.65	8.65	31.19	83.80	−0.138	Cytoplasm
PgCAD25	XM_031551278.1	359	38,947.64	6.59	28.04	91.78	−0.035	Cytoplasm

**Table 2 genes-14-00026-t002:** Estimated divergence period of the PgCAD gene pairs. Ks, synonymous substitution rate; Ka, non-synonymous substitution rate.

Gene Pair	Ka	Ks	Ka/Ks
PgCAD5-PgCAD8	0.280640639	3.285317089	0.085422695
PgCAD6-PgCAD	0.207931028	2.218153296	0.093740603
PgCAD7-PgCAD18	0.113573012	1.141785238	0.099469679

**Table 3 genes-14-00026-t003:** The four duplicated types of *PgCADs*.

Gene Name	Whole Genome Duplication or Segmental	Tandem Duplication	Dispersed Duplication	Proximal Duplication
PgCAD1			√	
PgCAD2				
PgCAD3			√	
PgCAD4			√	
PgCAD5	√			
PgCAD6	√			
PgCAD7	√			
PgCAD8	√			
PgCAD9		√		
PgCAD10		√		
PgCAD11		√		
PgCAD12		√		
PgCAD13				
PgCAD14		√		
PgCAD15		√		
PgCAD16				
PgCAD17			√	
PgCAD18	√			
PgCAD19			√	
PgCAD20				
PgCAD21		√		
PgCAD22		√		
PgCAD23				√
PgCAD24				√
PgCAD25				√

## Data Availability

The original transcriptome data used in this study has been submitted to the NCBI, and the data is stored in the SRA database, the acession number is “PRJNA914887”.

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
