# Peer review of "Identification and Functional Analysis of CAD Gene Family in Pomegranate (Punica granatum)"

_genes, 2022, doi:10.3390/genes14010026_

Round 1
Reviewer 1 Report
The manuscript entitled “Identification of the CAD Gene Family in Pomegranate and Functional Analysis” describes bioinformatic analysis of CAD gene family in pomegranate (Punica granatum) with the focus on homologs comparison, protein subcellular localization, phylogenetical and collinearity, and gene structure and protein conserved domains. The idea behind this manuscript is quite interesting but unfortunately the way manuscript has been provided has declined the overall quality of scientific outcome. Please refer to other similar works to have an overview how the data and information especially in material and method should be presented. Altogether the manuscript needs a major revision to be ready for further consideration. With comments:
1-The title should be revise. May it be better to write: Identification and functional analysis of CAD gene family in pomegranate (Punica granatum).
2- The abstract is somewhat null and no data value has been provided. You can some of the most important quantitative data to abstract. Moreover, it doesn’t meet Gene standards as it is too long and, some parts are without correct sentence structures. No need to bring information like the used software for analyses. Therefore, the abstract should be completely revised.
3- Altogether, the text needs a substantial English editing and revision as there many sections and sentences which need to be revised or rewritten. I have highlighted some examples throughout of the manuscript:
L8-10: In the first 3 lines, there is no verb.
L11: on the reported CAD → on the reported CAD sequence
It is better to refer the gene members of CAD family instead of CAD family throughout of the manuscript.
L17-18: Pomegranate CAD gene has 25 gene family members → Pomegranate CAD gene family has 25 members. In follow: “with similar physical and chemical properties”, you are speaking about gene sequence not protein. Please revise.
L20: “the members of group III may be unrelated to lignin” an incorrect conclusion.
L27-28: Please specify that Hongyushizi, Baiyushizi and Tunisia are cultivars.
L33: originated → has originated; moreover, Iran and Afghanistan are not part of Central Asia, but part of the Middle East.
L39: cultivar → cultivars
L45 and 53: “The formation of seed hardness characters of pomegranate” us reduandant and it could be written as “Development of hardiness in pomegranate seed”
L69: 9 CAD genes were → Nine CAD genes were
There are many other examples of grammatical and structural problem that I couldn’t point all. SO, I think the manuscript should go under an extensive polishing by a native expert in the field.
4- There are some very related works which have not cited in the present study. The:
Chao, N., Yu, T., Hou, C., Liu, L., & Zhang, L. (2021). Genome-wide analysis of the lignin toolbox for morus and the roles of lignin related genes in response to zinc stress. PeerJ, 9, e11964.
Li, Y., Wang, R., Pei, Y., Yu, W., Wu, W., Li, D., & Hu, Z. (2022). Phylogeny and functional characterization of the cinnamyl alcohol dehydrogenase gene family in Phryma leptostachya. International Journal of Biological Macromolecules, 217, 407-416.
Singh, S., Sharma, N., Malannavar, A. B., Badiyal, A., & Sharma, P. N. (2022). Cloning and in silico characterization of cinnamyl alcohol dehydrogenase gene involved in lignification of Tall fescue (Festuca arundinacea Schreb.). Molecular Genetics and Genomics, 297(2), 437-447.
Yusuf, C. Y. L., Nabilah, N. S., Taufik, N. A. A. M., Seman, I. A., & Abdullah, M. P. (2022). Genome-wide analysis of the CAD gene family reveals two bona fide CAD genes in oil palm. 3 Biotech, 12(7), 1-19.
Chao, N., Huang, S., Kang, X., Yidilisi, K., Dai, M., & Liu, L. (2022). Systematic functional characterization of cinnamyl alcohol dehydrogenase family members revealed their functional divergence in lignin biosynthesis and stress responses in mulberry. Plant Physiology and Biochemistry, 186, 145-156.
5- Please provide all exploited information in material and methods even default parameters of used software. This could include Gene ID of used sequence, gap penalty and gap extension used, exact version of software (in case of on-line tools mention access date) and other cases.
6- In material and methods little information has been provided including: RNA extraction and quality evaluation; cDNA synthesis. Moreover, the RNA seq information should be provided.
7- Table 3 could be bring in better and more compact way as there is no value for the last columns.
Author Response
Dear Reviewers,
Thank you for your letter and for the reviews’ comments concerning our manuscript entitled " Identification and functional analysis of CAD Gene Family in Pomegranate (Punica granatum)"(Genes-2040501). Those comments are all valuable and very helpful for revising and improving our paper. We have studied comments carefully and have made correction which we hope meet with approval. The main corrections in the paper and the responds to the reviewer’s comments are in the word.
Yours Sincerely,
Lei Hu
Corresponding author:
Name: Shuiming Zhang

Reviewer 2 Report
The paper by Hu et al. entitled 'Identification of the CAD Gene Family in Pomegranate and Functional Analysis' is dealing with an important topic which is the explanation of hard- and soft seed in pomegranate
the paper analysed in details this subject but I have many comment to the authors in order to further improve the quality of the paper
The abstract is fine but relatively long and many details should be deleted and simlified
The introduction should be improved in my oipnion authors should clearly state what has been achieved on the topic on the paper previously and what is lacking for pomegranate to be done
The materials and methods section is ok
Results section is fine but try to furher improve simplify avoid unnecessary data and correct the mistakes
Discussion Some mistakes should be corrected and sentences improved
Conclusion section is really poor and needs a special care by the authors
suggestions for duture research should be at least highlighted
******************
other comments and suggestions are attached

Author Response
Dear Reviewers,
Thank you for your letter and for the reviews’ comments concerning our manuscript entitled " Identification and functional analysis of CAD Gene Family in Pomegranate (Punica granatum) "(Genes-2040501). Those comments are all valuable and very helpful for revising and improving our paper. We have studied comments carefully and have made correction which we hope meet with approval. The main corrections in the paper and the responds to the reviewer’s comments are in the word.
Yours Sincerely,
Lei Hu
Corresponding author:
Name: Shuiming Zhang

Round 2
Reviewer 1 Report
Dear authors
Thanks for the substantial revision made to the manuscript. Please double-check all sections of the manuscript. The manuscript can go for further evaluation.
Reviewer 2 Report
The authors have done an intensive editing to the first version
I think now that the paper is suitable for publication